# Stakeholders' attitudes to outdoor residual spraying technique for dengue control in Malaysia: A PLS-SEM approach

Ahmad Firdhaus Arham [1]*, Latifah Amin[1,2]*, Muhammad Adzran Che Mustapa[1], Zurina Mahadi[1], Mashitoh Yaacob[1], Maznah Ibrahim[1]

**1** Pusat Pengajian Citra Universiti, Universiti Kebangsaan Malaysia, UKM Bangi, Selangor, Malaysia, **2** The Institute of Islam Hadhari (HADHARI), Universiti Kebangsaan Malaysia, UKM Bangi, Selangor, Malaysia

* benferdaoz@ukm.edu.my (AFA); nilam@ukm.edu.my (LA)

**Data Availability Statement:** All relevant data are within the manuscript.

## Abstract

Outdoor Residual Spraying (ORS) technique is a complementary preventive measure for dengue. The alarming number of dengue cases in Malaysia requires an alternative method to control dengue besides the traditional method such as fogging. However, the introduction of new technologies depends on social acceptance. Therefore, this study was important to determine the factors that influence stakeholders' attitudes towards the ORS and the moderating factor. A validated instrument was used to randomly interview 399 respondents representing two stakeholder groups which consist of scientists, and the public in Klang Valley, Malaysia. The findings revealed that the stakeholders claimed to have a high degree of religiosity, a high level of trust in the key players, perceived ORS as having high benefits, and displayed highly positive attitudes towards the ORS. The attitudes model towards the ORS model was developed using the SmartPLS software version. The perceived benefit was endorsed as the most important direct predictor of attitudes towards the ORS (ß = 0.618, P<0.001), followed by trust in the key players (ß = 0.151, P<0.001). It is also interesting to note that religiosity served as a moderator for the association between perceived benefit (ß = 0.075, P = 0.024) and perceived risk (ß = 0.114, P = 0.006) with attitudes towards the ORS. The identified predictor factors of stakeholders' attitudes toward the ORS and the moderating factor can serve as indicators for social acceptance of ORS in developing countries. These indicators can help the policymakers in decision making to implement this technique.

## Author summary

The ups and downs of dengue cases require the Outdoor Residual Spraying (ORS) technique as an alternative method to control dengue in Malaysia. However, the introduction of ORS depends on stakeholders' acceptance. Here, the purpose of this study was to determine the factors that predict attitude to ORS and the role of religiosity as a moderator. The results indicate positive responses for implementing the ORS as a suitable technique to control dengue in Malaysia. Perceived benefit emerged as the most significant direct

**Funding:** This research was supported by a research grant under the post-doctoral scheme RGA1 and MI-2020-010 from Universiti Kebangsaan Malaysia and the Ministry of Higher Education, Malaysia under the ERGS/1/2013/SSI12/UKM/02/1 grant. The funders had no role in study design, data collection and analysis, decision to publish, or preparation of the manuscript.

**Competing interests:** The authors have declared that no competing interests exist.

predictor of attitude towards the ORS ($\beta = 0.618$, $p < 0.001$), which they viewed this technique as less risky. Trust in key players had a significant positive relationship with attitudes towards the ORS that makes this factor the second most important direct predictor ($\beta = 0.151$, $p < 0.001$). Interestingly, the religiosity factor significantly moderated the relationship between perceived benefit and risk with attitudes towards the ORS. This study also showed the role played by both perceived benefit and risk as mediating factors.

## Introduction

Dengue fever is a debilitating *Aedes* mosquito-borne disease that has spread throughout the world, including Malaysia [1,2]. The increase in dengue cases is alarming, as 100 million cases of classical dengue fever and 500,000 cases of dengue haemorrhagic fever are annually recorded worldwide [3,4]. The highest dengue fever cases were reported in 2015 at 120,836 cases with 336 deaths [5]. In 2019, the official website of iDengue for community by the Ministry of Health (MOH), Malaysia and Ministry of Science, Technology, and Innovation (MOSTI), Malaysia (http://idengue.arsm.gov.my) reported a decrease of cases from the previous year hitting nearly 80,000 cases, with 152 deaths from January until October. Although the number of cases has declined, it remains a major concern [5].

In Malaysia, more than 70% of dengue cases are reported in rapidly developing urban areas with the highest density population [1,3]. Increases in population, urbanization, globalization, global warming, concentration of social and environmental processes, climate change and lack of control of vectors have contributed to the growth in the number of dengue cases [3,6–12]. *Aedes aegypti* mosquito is a main vector of dengue fever and caused by four related dengue virus serotypes—DENV-1, DENV-2, DENV-3, and DENV-4, which do not offer long-term immunity protection against one another. *Aedes* mosquitoes are easily reproduced in water reservoir, either outdoors or in the house. These mosquitoes breed easily in water storage containers around the human habitat [13], especially in clear, clean and calm reservoirs such as flower baskets, buckets, plastic containers, water tanks, old tires, coconut shells, clogged drains, and so on [14,15].

The fogging technique is the main technique used to control and prevent dengue fever in Malaysia. However, this conventional technique and public awareness against dengue is not efficient enough to combat dengue [16]. Therefore, various alternative control measures have been introduced including the residual spraying technique as an insecticide-based control. Laura et al. (1998), highlighted that indoor and outdoor residual spraying is a method frequently used to reduce vector-man contact [17]. Rozilawati et al. (2005), explained that house residual spraying continues to be an effective prevention tool for malaria in tropical countries. It is important to control the vector population, and residual spraying can provide effective control in specific areas because effective dengue vaccines and antiviral drugs are still not available [18]. Earlier studies have reported that residual spraying technique is effective in controlling *Aedes*, especially in highly populated areas [18–24]. The Outdoor Residual Spraying (ORS) technique is a complementary method that involves spraying the outer wall of a house, premises, or any buildings. However, the effectiveness of this technique depends on the features of the wall surface, the geographical area, rainfall, humidity, and temperature.

Rozilawati et al. (2005) evaluated the ORS using traditional deltamethrin formulated as water dispersible granule in an urban residential area consisted of a block of flats and single-storey wood-brick houses in Kuala Lumpur [18]. The flats were used as control groups while the houses were treated with the ORS. Based on biweekly bioassay results, residual spraying of

deltamethrin was still effective for six weeks after treatment and indicated that both *Aedes aegypti* and *Aedes albopictus* were more susceptible on the wooden surfaces than on the brick. *Aedes aegypti* was more susceptible than *Aedes albopictus* against deltamethrin. In this study, residual spraying of deltamethrin was ineffective since the *Aedes* population area did not reduce as indicated by the total number of larvae collected using the ovitrap. Several reasons for the field ineffectiveness of the ORS include change of resting behaviour of *Aedes*, invasion of *Aedes* from nearby areas, or use of ovitrap surveillance may not be a good indicator [19]. Subsequently, a further analysis was performed by Hamid et al. (2019), using modern polymer deltamethrin developed as a suspension concentrate (SC-PE 30) in low-and high-rise residential areas. Bioassay findings revealed that the *Aedes* mortality rate was more than 80 percent for 16 weeks [23].

A recent research by the Institute of Medical Research (IMR) Malaysia found that the ORS technique is workable and more effective to serve as a complementary tool to the current space spraying such as fogging technique that is regularly used during dengue outbreaks in Malaysia [24]. Field studies were conducted by the IMR to assess the effectiveness of the ORS on the outside of a five-story apartment building using spray-formulated pyrethroids containing deltamethrin-coated polymer versus another apartment building in the same area treated with conventional dengue control measures as a control method [2]. The efficacy of this technique was tested within a year by observing the effect of vector population density by using ovitrapping methods and dengue epidemiological data [2]. The results found that the ovitrap index was reduced by 50% compared to the control sites after one-month spraying, thus lowering the monthly incidence of dengue fever cases, with no dengue cases being reported throughout the study. Hamid et al. (2020), further compared the residual bio-efficiency of modern deltamethrin SC-PE with traditional deltamethrin WG on treated cement surfaces introduced to the outer walls by the IMR. Analyses of the bioassay results showed that modern deltamethrin SC-PE increased longevity to week 17 relative to traditional deltamethrin, which only performed until week 10. Post-hoc test results found that the modern deltamethrin SC-PE had the highest mortality rate in *Aedes* mosquitoes [24].

As mentioned earlier, the ORS technique has good potential as an effective alternative technique for controlling dengue disease. However, the effectiveness of this technique is affected by environmental factors that involve weather aspects such as the rate of rainfall or the amount of sunlight, and the type of premises treated with this method [2]. A comprehensive strategy for a cost-effective and efficient implementation of the ORS technique needs the combined efforts of the government, the private sectors, industry, and society. Therefore, we hypothesized that the large-scale introduction of the ORS into the environment tends to rely on stakeholder's attitudes. The verification of this research hypothesis could contribute to the acceptance and implementation of the ORS. However, until now there has been no documented study on attitudes towards the ORS in Malaysia. Hence, the purpose of this study was to determine the attitude and factors that predict attitude to ORS and the role of religiosity as a moderator.

## Theoretical framework and hypotheses development

The theoretical framework of the study was developed from the work of Amin and Hashim (2015) [16], which was based on Fishbein's Multi-Attribute Model (1963) [25]. The model begins with a listing of predictive factors that affect the attitudes towards the ORS, perceived benefit, perceived risk, attitude to nature versus materials, trust in key players, and religiosity as a moderating factor.

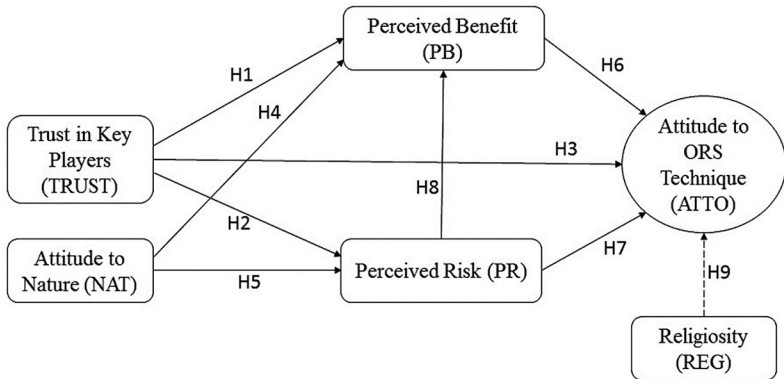

**Fig 1. Research framework for predicting factors and religiosity as a moderator in understanding the attitudes of stakeholder toward the ORS technique.**

Attitude to ORS technique is the endogenous variable for the four exogenous variables: perceived benefits, perceived risks, trust in key players and attitude to nature versus materials. Perceived benefits and perceived risks acted as the mediating variables for the relationships between the variables: trust in key players and attitude to nature versus materials and attitude to ORS technique. Additionally, religiosity was proposed as the moderating factor, which affect the strength of the relationship between the exogenous variables and attitude to ORS technique.

To assess the relevance in association among these variables, the component-based approach of the PLS-SEM (Partial Least Squares Structural Equation Modeling) was used with SmartPLS software. Fig 1 presents the conceptual research framework for the proposed relationships between the exogenous and endogenous variables, as well the role of the mediating and moderating variables. The hypotheses were developed based on the correlations results between the factors using the Pearson correlation method [26] (Table 1). The component-based approach of PLS-SEM (Partial Least Squares Structural Equation Modeling) of the SmartPLS software was used to test the hypotheses.

## Trust in key players

Public attitudes towards new technologies depend on their trust in key players, which are seen as experts or responsible institutions from within the industry. Trust in key players is an important predictor of public acceptance of modern technology [27–29]. Limitations in terms of knowledge cause the public to rely on experts when it comes to accepting modern technology [30]. The community does not directly assess the benefits and risks associated with

**Table 1. The correlation matrix among the factors in the research (ORS technique).**

| | | | | | | |
|---|---|---|---|---|---|---|
| [1]Trust in Key Players | - | | | | | |
| [2]Attitude to Nature versus Materials | -0.019 | - | . | | | |
| [3]Perceived Benefit | 0.408** | 0.131** | - | | | |
| [4]Perceived Risk | -0.182** | -0.225** | -0.209** | - | | |
| [5]Attitude to ORS Technique | 0.405** | 0.043 | 0.678** | -0.182** | - | |
| [6]Religiosity | 0.155** | -0.224** | 0.180** | -0.046 | 0.167** | - |

Note: * $p < 0.05$

** $p < 0.01$.

genetically modified (GM) products, rather relies on information provided by experts in the industry [16]. Gaskell et al. (2003) suggested that the acceptance of GM food relies on public trust in the government, industry, and retailers [31]. Meanwhile, Chen and Li (2007) found that consumer trust depends on their beliefs on the science and institutions involved in genetic technology, to determine the associated benefits and risks [32]. In fact, the study by Bronfman et al. (2009), highlighted that when the public's trust in the regulatory institutions is so positive, they would even accept any harm resulting from the technology [33]. Consequently, the following hypotheses are proposed:

*H1*: *The more trust in the key players involved in using or applying the ORS, the more benefits will be perceived of the ORS.*

*H2*: *The more trust in the key players involved in using or applying the ORS, the fewer risks will be perceived of the ORS.*

*H3*: *The more trust in the key players involved in using or applying the ORS, the more relatively positive attitude towards the ORS.*

## Attitude to nature versus materials

Attitude to nature refers to the inclination of the respondents, either to preserve nature or to place an emphasis on man-made materials [34]. Past studies mentioned that this factor is referred to as nature and societal value [34,35]. People who placed materials above nature value tended to support modern technology [16]. Therefore, the researchers have proposed the following hypotheses to clarify the association between attitudes to nature versus materials with the stakeholders' attitudes towards the ORS:

*H4*: *The more inclination towards materials than nature, the more benefits will be perceived of the ORS.*

*H5*: *The more inclination towards materials than nature, the fewer risks will be perceived of the ORS.*

## Perceived benefit and risk

Studies on consumer behaviour have indicated that perceived benefit and risk strongly influenced individual responses regarding acceptance of technology [36–41]. Perceived benefit is a positive predictor of attitude, while perceived risk is a negative predictor of attitude. Perceived risk refers to the loss that an individual will suffer rather than an unfavourable activity, thing, or matter [42]. Perceived benefit and risk are difficult to conceptualize separately because of their complex association which has an inverse relationship [43–45]. Past studies have found that people tend to support any biotechnology applications if the perceived benefit outweighs the risk [40]. Therefore, the following hypotheses are proposed:

*H6*: *The more benefits associated with the ORS, the more positive attitude towards the ORS.*

*H7*: *The more risks associated with the ORS, the more negative attitude towards the ORS.*

*H8*: *The more risks associated with the ORS, the lower benefits will be associated with the ORS.*

## Religiosity

Religion is an important element that may influence a person's attitude and opinion regarding the acceptance of new technologies [46]. The influence of religious commitment has been

widely studied in psychological research [47,48]. Amin et al. (2011), reported that religious people tend to be more critical of modern technology, and they perceive GM products as being risky, despite acknowledging their benefits [49]. Stakeholders in Malaysia have been reported as having high attachment to their religion [38,50,51]. Therefore, the researcher examines religiosity as a moderating factor in assessing public's attitude towards the ORS. This factor moderates the relationship between predictive factors that directly relate to attitude towards the ORS. The following hypotheses are proposed:

**H9**: *Religiosity moderates the relationship between perceived benefit and attitude towards the ORS.*

**H10**: *Religiosity moderates the relationship between perceived risk and attitude towards the ORS.*

**H11**: *Religiosity moderates the relationship between trust in key players and the attitude towards the ORS.*

## Methodology

This survey was conducted among 415 Malaysian adults (aged 18 years and above) in the Klang Valley region, who were chosen using the stratified random sampling technique. Initially the respondents were stratified according to two categories which are the scientists and the public followed by simple random sampling selection. A total of 399 out of 415 questionnaires were analysed whereby 16 questionnaires were rejected due to incomplete responses and bias. The data was collected between September 2016 to September 2017. The Klang Valley population was chosen as a target population as it is the centre for socio-economic development in Malaysia. At the same time, the Klang Valley, which covers the state of Selangor and Kuala Lumpur, has been identified as having the highest incidence of dengue fever cases according to the Malaysian Ministry of Health (http://idengue.arsm.gov.my).

The G*Power 3.1.9.2 software was used to calculate statistical power of 0.80 and the minimum sample size for this study [52,53]. The effect of the size ($f$ = 0.15) and the significance level ($p < 0.05$) meant that a minimum total sample size of 277 respondents was required according to G*Power calculation. In this study, the respondents were split into two groups: the public (n = 197) and the scientists (n = 202). Respondents representing the public were selected from the community living in outbreak areas, identified as having high densities of *Aedes* mosquitoes in the Klang Valley. The scientists included academics, postgraduate student, and research officers involved in environmental science, biological sciences, health and genetic sciences research as well as who were involved in the control and prevention of dengue disease. The combination of these two stakeholders is important as they have similar interests as major potential beneficiaries of the ORS technique.

No ethical approval was required in this study according to the Guidelines for the Ethical Review of Clinical Research or Research involving human subjects by Medical Review and Ethics Committee (MREC), Ministry of Health Malaysia [54]. The research would be exempt from MREC approval because this study involving the use of questionnaires to explore public behaviour with no collection of identifiable private information. MREC also may waive the requirement to obtain individual informed consent if subjects are exposed to no more than minimal risk and the study involves only publicly available data. However, informed consent was obtained verbally from all respondents before they answer the questionnaire. The participation of the respondents was voluntary, and withdrawals were allowed at any time. If they agree to answer this questionnaire, only the respondent's identification number and the date the questionnaire is answered are written on the front of the survey paper without personal

information. The respondents answered this questionnaire face to face with the researchers and enumerators involved to ensure all questions were answered directly at that time.

The multi-dimensional survey instrument assessing stakeholders' attitudes towards the ORS technique was constructed on the basis of previously published research [16]. The instrument consisted of six variables namely attitude, perceived benefits, perceived risks, trust in key players and attitude to nature versus material and religiosity as a moderator (Fig 1). All items were measured using a 7-point Likert scale, ranging from 1 (strongly disagree) to 7 (strongly agree).

The Statistical Package for Social Sciences (SPSS) version 24 software was used to analyse the descriptive and inferential statistics, the reliability analysis, and the internal consistency of the variables. SmartPLS software version 3.2.7 examined the structural model to determine the relationship between the variables by performing a bootstrapping with 5000 resamples. All table and figure files are available from the https://figshare.com/account/home#/projects/113760database.

## Results and discussion

Table 2 presents the overall mean scores of stakeholders' attitudes towards the ORS and its predicting factors. For ease of interpretation, the overall mean scores for all the variables have been categorized into three categories. If the mean scores were within the range of 1 to 3.00, they would be categorized as low. When the mean scores were in the range of 3.01 to 5.00, they would be classified as moderate, while mean scores of 5.01 to 7.00 were categorized as high. The overall mean scores for religiosity (mean score of 6.11), trust on key players (mean score of 5.51) and perceived benefits (mean score of 5.27) were in the high category. The stakeholders claimed that they were highly attached to their religion, were highly trusting of the key players and perceived the ORS technique as highly beneficial which was translated into a highly positive attitude to ORS technique (mean score of 5.28). On the other hand, the stakeholders were more inclined towards nature (mean score value were 3.92, below the mid-point of 4.0). Additionally, the stakeholders perceived the risks of the ORS technique as moderate (mean score of 3.30).

### Analysis of the measurement model

Four evaluation criteria: factor loadings, composite reliability, convergence validity, and discrimination validity were used to measure the validity and reliability of the constructs [55]. Table 3 shows that all the factor loadings were above the value of 0.5, and the total average variance extracted (AVE) exceeds 0.5, which are considered acceptable [56,57]. The composite reliability (CR) values for all variables reached the minimum value of 0.7 as recommended by Hair et al., (2014) and Gefen et al. (2000) [55,58].

**Table 2. Mean Score and Standard Deviation.**

| Factor | Mean ± Standard Deviation | Interpretation |
|---|---|---|
| Trust in Key Players | 5.51 ± 0.94 | High |
| Attitude to Nature versus Materials | 3.92 ± 1.49 | Moderate |
| Perceived Benefit | 5.27 ± 1.13 | High |
| Perceived Risk | 3.30 ± 1.20 | Moderate |
| Attitude to ORS Technique | 5.28 ± 1.06 | High |
| Religiosity | 6.11 ± 1.10 | High |

Note: 1–3.00, low; 3.01–5.00, moderate; 5.01–7.00, high.

**Table 3. Internal Consistency and Convergent Validity.**

| Item | Factor Loadings | CR | AVE | Validity |
|---|---|---|---|---|
| **Trust in Key Players** | | 0.878 | 0.705 | YES |
| TRUST 1: Scientists have done a good job for society. | 0.846 | | | |
| TRUST 2: Industries have done a good job for society. | 0.847 | | | |
| TRUST 3: Government have done a good job for society. | 0.826 | | | |
| **Attitude to Nature versus Materials** | | 0.902 | 0.698 | YES |
| NAT 1: Society aiming to preserve nature versus society stressing to achieve wealth. | 0.758 | | | |
| NAT 2: Society with a centrally planned economy versus society relying on a market-driven economy. | 0.838 | | | |
| NAT 3: Society that will stop development at the expense of any risks versus society that deliberately accepting any risks for the attainment of wealth. | 0.899 | | | |
| NAT 4: Society that optimizes the protection of the environment above economic growth versus society that stress nature can with stand human actions. | 0.841 | | | |
| **Perceived Benefit** | | 0.923 | 0.632 | YES |
| PB 1: ORS technique will enhance the quality of life. | 0.767 | | | |
| PB 2: ORS technique is useful to the Malaysian society. | 0.794 | | | |
| PB 3: ORS technique is useful in preventing dengue fever. | 0.838 | | | |
| PB 4: ORS technique is effective to eradicate dengue. | 0.848 | | | |
| PB 5: ORS technique is beneficial to me and my family's health. | 0.796 | | | |
| PB 6: The benefits of the ORS technique to people outweigh their risks. | 0.758 | | | |
| PB 7: Whatever the risks of the ORS technique will be dealt with future research. | 0.756 | | | |
| **Perceived Risk** | | 0.938 | 0.685 | YES |
| PR 1: Level of worries about the unknown effects of the ORS technique? | 0.688 | | | |
| PR 2: Any harmful effects from using the ORS technique will only manifest itself after long term duration? | 0.826 | | | |
| PR 3: Using the ORS technique will pose threat to future generation. | 0.856 | | | |
| PR 4: ORS technique may give rise to unknown consequences. | 0.888 | | | |
| PR 5: Any danger from the ORS technique may cause a major catastrophe to Malaysian society. | 0.840 | | | |
| PR 6: How worried are you about the potential risks of the ORS technique to your health and you family's health? | 0.813 | | | |
| PR 7: Adverse effects from the ORS technique are harmful. | 0.869 | | | |
| **Attitude to ORS Technique** | | 0.879 | 0.594 | YES |
| ATO 1: ORS technique should be scaled up. | 0.731 | | | |
| ATO 2: Government should provide more financial support to researchers and industries in developing the ORS technique. | 0.692 | | | |
| ATO 3: ORS technique help government to decrease community's mortality. | 0.725 | | | |
| ATO 4: ORS technique is necessary. | 0.828 | | | |
| ATO 5: ORS technique is encouraged. | 0.862 | | | |
| **Religiosity** | | 0.946 | 0.815 | YES |
| REG 1: Religion is important in my life. | 0.922 | | | |
| REG 2: Religious views are important when I have to make decisions about controversial issues. | 0.902 | | | |
| REG 3: Praying is important in my life. | 0.918 | | | |
| REG 4: Reading scriptures is important in my life. | 0.868 | | | |

Note: Factor loadings, internal consistency of the items loading; Composite Reliability (CR), square of the summation of the factor loadings; Average Variance Extracted (AVE), summation of the square of the factor loadings.

Discriminant validity is considered a prerequisite for analysing relationships between latent variables [59]. In this study, two types of discriminant validity determined are i) Fornell-Larcker Criterion and ii) Heterotrait-Monotrait Ratio (HTMT). In the Fornell-Larcker Criterion assessment, each of the constructs had a higher square root value of AVE than the correlation estimates of the constructs, thus considered acceptable, as suggested by Chin (2010) [60] (Table 4). Meanwhile, the value of the $HTMT_{0.90}$ correlation for each of the constructs was less than 0.85 and was rated as acceptable as suggested by Kline (2015) [61] (Table 5).

## Analysis of the structural model

The evaluation of collinearity was done before the analysis of the structural model to ensure that there were no collinearity concerns regarding the inner model of the study. The result of the collinearity test of the inner model showed that the variance inflation factor (VIF) values

**Table 4. Fornell-Larcker Criterion Correlation.**

| Factor | 1 | 2 | 3 | 4 | 5 | 6 |
|---|---|---|---|---|---|---|
| [1] Trust in Key Players | **0.840** | | | | | |
| [2] Attitude to Nature versus Materials | -0.014 | **0.835** | | | | |
| [3] Perceived Benefit | 0.416 | 0.124 | **0.795** | | | |
| [4] Perceived Risk | -0.197 | -0.257 | -0.237 | **0.828** | | |
| [5] Attitude to ORS Technique | 0.420 | 0.044 | 0.706 | -0.215 | **0.770** | |
| [6] Religiosity | 0.154 | -0.218 | 0.182 | -0.065 | 0.172 | **0.903** |

Note: The results had a square root of AVE value that exceeded the total variance shared with another variable factors.

were below 5.0 for each of the constructs, as suggested by Hair et al. (2014) [55]. Before hypotheses testing, measurement of the model fit was carried out [62]. The value of the standardized root mean square residual (SRMR) was 0.069 ($< 0.08$), which was considered a good fit for PLS path models [63].

The analysis of the structural model also involved, i) testing for the co-efficient of determination ($R^2$), ii) testing for the effect size ($f^2$) on the impact value of the exogenous variables on the endogenous variable, and iii) testing for the predictive accuracy of the model predictions ($Q^2$) (Table 6). A well-fitting model should have $R^2$ in the range from 0 to 1. The exogenous variables in the model were able to explain 52.9% of variance in attitude towards the ORS approach. The $R^2$ value of the perceived benefit factor was 0.207, which suggested that the exogenous variables have explained 20.7% of the factor. While the R2 value for the perceived risk was 0.106 (10.6% variance).

According to Cohen (1988) [64], perceived benefit ($f^2 = 0.615$) has a large effect size on attitude towards the ORS approach compared with trust in key players ($f^2 = 0.038$). At the same time, trust in key players has a medium effect size on the perceived benefit factor ($f^2 = 0.184$), while the effect size of perceived risk and attitude to nature versus materials on perceived benefit was small ($f^2 = 0.021$). On the other hand, trust in key players ($f^2 = 0.045$) and attitude to nature versus materials ($f^2 = 0.075$) have a small effect size on the perceived risk.

The $Q^2$ values for the perceived benefit (0.120), perceived risk (0.069), and attitudes towards the ORS approach (0.287) were greater than zero which confirmed that the model's predictive relevance was adequate for the exogenous variables [59, 65].

## Relationship among the constructs

Eight hypotheses (H1 to H8) postulated the direct association between the predictors and attitudes towards the ORS approach, while three hypotheses (H9, H10, H11) tested the

**Table 5. HTMT Ratio Correlation.**

| Factor | 1 | 2 | 3 | 4 | 5 | 6 |
|---|---|---|---|---|---|---|
| [1] Trust in Key Players | | | | | | |
| [2] Attitude to Nature versus Materials | 0.109 | | | | | |
| [3] Perceived Benefit | 0.484 | 0.152 | | | | |
| [4] Perceived Risk | 0.230 | 0.274 | 0.246 | | | |
| [5] Attitude to ORS Technique | 0.501 | 0.116 | 0.790 | 0.232 | | |
| [6] Religiosity | 0.216 | 0.248 | 0.198 | 0.086 | 0.192 | |

Note: The values of $HTMT_{0.90}$ does not exceed 1 that means the indicator for that factor is less than the discriminant validity aspect.

**Table 6. Determination of Co-efficient ($R^2$), Effect size ($f^2$) and Predictive Relevance ($Q^2$).**

| | $R^2$ | $Q^2$ | Effect Size ($f^2$) | | | |
|---|---|---|---|---|---|---|
| | | | Perceived Benefit | Perceived Risk | Trust in Key Players | Attitude to Nature versus Material |
| Attitude to ORS Technique | 0.529 | 0.287 | 0.615 (Large) | | 0.038 (Small) | |
| Perceived Benefit | 0.207 | 0.120 | | 0.021 (Small) | 0.184 (Medium) | 0.021 (Small) |
| Perceived Risk | 0.106 | 0.069 | | | 0.045 (Small) | 0.075 (Small) |

Note: $R^2$, range from 0 to 1; $f^2$, large $\geq$ 0.35, medium $\geq$ 0.15, small $\geq$ 0.02; $Q^2$, greater than 0.

moderating effect of religiosity (Fig 2). Table 7 illustrates the bootstrapping results using sub-samples of 5,000 cases to examine the relationship among the constructs. All the hypotheses were analysed simultaneously.

The results revealed that perceived benefit emerged as the most significant direct predictor of attitude towards the ORS ($\beta$ = 0.618, p<0.001) (Fig 2), suggesting that when the respondents in the Klang Valley perceived the ORS as having higher benefits, they would be more positive towards the technique. This finding is supported by previous studies where perceived benefit emerged as the main factor of attitudes to GM mosquito [16], and biodiesel products [37]. However, in this study, the perceived risk did not have any association with attitudes but had a negative association with the perceived benefit of ORS ($\beta$ = -0.136, p<0.01) (Fig 2). The findings indicated that when the respondents assessed the ORS as having higher benefits, they would view the technique as less risky. The result is in line with Amin et al. (2017), who reported an inverse relationship between perceived benefits and perceived risks of biodiesel [37]. Attitude have been reported to be determined by people's perceptions of benefits and risks associated with specific applications or technologies [66].

The findings of this study also highlighted that attitude to ORS involved the interplay between other factors such as trust in key players and attitude to nature versus materials. There was a significant positive relationship between trust in key players and attitudes towards the ORS that makes this factor the second most important direct predictor ($\beta$ = 0.151, p<0.001) (Table 7 and Fig 2). The results suggest that when stakeholders have a high level of trust in the key players involved in controlling dengue, they render positive attitudes towards the ORS. The study by Arham et al. (2018), also found a positive association between trusts in key players with the attitudes towards dengue prevention techniques [67]. Past studies have

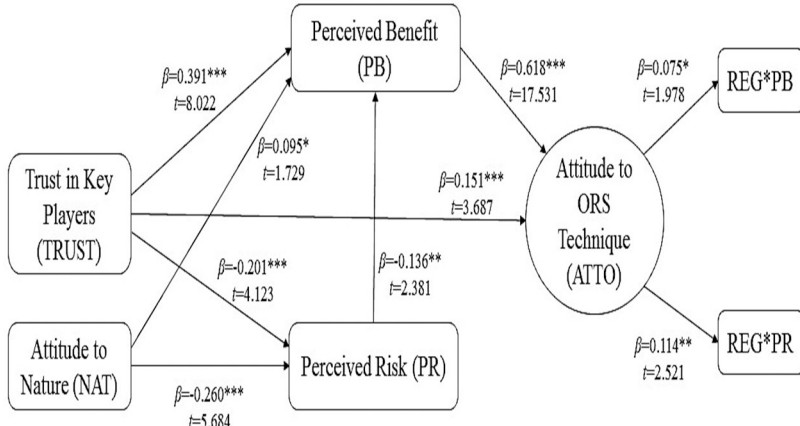

**Fig 2. Results of the structural model analysis for predicting factors and religiosity as a moderator in understanding the attitudes of stakeholders towards the ORS technique.**

**Table 7. The relationship predicting factors and moderator that influence attitude towards the ORS technique.**

| | Hypothesized Path | Path Coefficient | Standard Error | T-Values | P-Values | Decision |
|---|---|---|---|---|---|---|
| H1 | Trust in Key Players -> Perceived Benefit | 0.391 | 0.049 | 8.022 | 0.000*** | Supported |
| H2 | Trust in Key Players -> Perceived Risk | -0.201 | 0.049 | 4.123 | 0.000*** | Supported |
| H3 | Trust in Key Players -> Attitude to ORS Technique | 0.151 | 0.041 | 3.687 | 0.000*** | Supported |
| H4 | Attitude to Nature -> Perceived Benefit | 0.095 | 0.055 | 1.729 | 0.042* | Supported |
| H5 | Attitude to Nature -> Perceived Risk | -0.260 | 0.046 | 5.684 | 0.000*** | Supported |
| H6 | Perceived Benefit -> Attitude to ORS Technique | 0.618 | 0.035 | 17.531 | 0.000*** | Supported |
| H7 | Perceived Risk -> Attitude to ORS Technique | -0.053 | 0.042 | 1.278 | 0.101 | Not Supported |
| H8 | Perceived Risk -> Perceived Benefit | -0.136 | 0.057 | 2.381 | 0.009** | Supported |
| H9 | Religiosity*Perceived Benefit -> Attitude to ORS Technique | 0.075 | 0.038 | 1.978 | 0.024* | Supported |
| H10 | Religiosity*Perceived Risk -> Attitude to ORS Technique | 0.114 | 0.045 | 2.521 | 0.006** | Supported |
| H11 | Religiosity*Trust in Key Players -> Attitude to ORS Technique | 0.000 | 0.036 | 0.001 | 0.500 | Not Supported |

Note: **$p < 0.01$

*$p < 0.05$ (one-tailed).

cited that people's perception of the controlling body of a certain risk often affects their assessment of that particular risk [68]. Montgomery et al. (2010) stated that trust and involvement of key players especially local government leaders has led acceptance of residual spraying to control Malaria in Mozambique [69]. Therefore, when people have low confidence in the key actors, they tend to exaggerate the associated risks [31]. Trust in key players also have a positive relationship with perceived benefit (β = 0.391, p<0.001) and a negative relationship with perceived risk (β = -0.201, p<0.001). Fig 2 shows that when the respondents were more trusting towards the key actors, they viewed the ORS as having higher benefit and low risk.

Attitude to nature versus materials has a positive association with perceived benefit (β = 0.095, p<0.05) and a negative association with perceived risk (β = -0.260, p<0.001) (Fig 2). This explains that the respondents who were more inclined towards materials tend to perceive the ORS as being more beneficial and less risky. Previous findings also demonstrated that the respondents who were more materialistic perceived fewer risks associated with the development of xenotransplantation [36].

It is also an interesting finding that religiosity significantly moderated the relationship between perceived benefit and risk with attitudes towards the ORS in this study. This indicated that when the stakeholders have a high religious commitment, the strength of the relationship between the perceived benefit with attitudes towards the ORS was weaker (β = 0.075, p<0.05) (Fig 2). The religiosity factor has reduced the influence of perceived benefit on attitudes towards the ORS. Consequently, the influence of perceived benefit on attitudes towards this technique is higher when the degree of religiosity is less. Additionally, religiosity also moderated the relationship between perceived risk and attitudes towards the ORS (β = 0.114, p<0.01). This suggests that the association is weakened when religious commitment is higher. In other words, religious commitment weakens the strength of the relationship between perceived risk and attitudes towards the ORS. A possible explanation is that when people are more religious, they become more cautious. Previous findings reported that religiosity has a positive association with both the general promise of modern biotechnology and the perceived risk of agro-biotechnology [49]. Therefore, there is a need for a more in-depth study to understand the complex role of religiosity as a moderator for attitude to new technology.

The results of this study also showed the role played by both perceived benefit and risk as mediating factors. The variance accounted for (VAF) value has been calculated to measure the

size of the indirect effects of the mediator variable on the path from exogenous to endogenous variables. It was determined by dividing the direct effect with the total effect as suggested by Zhao et al. (2010), and if the value exceeds 20%, it indicates the mediating effect [70].

Perceived benefit has mediated the relationship between trust in key players (Trust in Key Players>Perceived Benefit>Attitudes towards the ORS = 0.151, t = 3.687, p = 0.000) (Table 7 and Fig 2) at a 95% confidence level (p<0.05) and the attitudes towards the ORS with VAF value of 61.54%. The result confirmed that perceived benefit acted as a mediator, indicating that when the stakeholders have high trust in the key players, the ORS was perceived as highly beneficial, yielding a positive attitude towards the technique.

Furthermore, the perceived risk also served as a mediator (Attitude to Nature versus Materials>Perceived Risk>Perceived Benefit = 0.095, t = 1.729, p = 0.042) (Table 7 and Fig 2) at a 95% confidence level (p<0.05) with the VAF value at 27.12%. This finding demonstrated that when the respondent who was more inclined towards materials, they tended to assess the risk of this technique as being lower, and they endorsed a higher perceived benefit of the ORS.

There are several limitations to this study related to the study area and samples. This study concentrated on the dengue hot spots areas in the Klang Valley. Future studies can be extended to other areas throughout Malaysia as well as throughout Asia. Wider coverage in terms of research areas will present a more diverse socio-cultural background and may give different results. Focusing the sampling only on the public and the scientists were also regarded as a limitation of the study. Sampling can be extended to other stakeholders such as the industry, the media, NGOs, and others, to see any differences in public attitudes towards the acceptance of this technique. There is also a need to study public attitudes to other dengue control techniques such as the Bacillus thuringiensis, the Wolbachia, the vaccine, and other alternatives to control the dengue disease.

## Conclusion

In conclusion, this research has confirmed that social acceptance or specifically the attitudes towards the ORS should be seen as a multi-faceted process. The results indicate positive responses for the implementation of the ORS as a suitable technique to control dengue in Malaysia. It has enormous potential for improving the quality of public health. The findings have demonstrated that the respondents were highly positive concerning the ORS because of its benefits and trust in key players. Notably is the role of religiosity as a moderator for the influence of perceived benefit and risk on stakeholders' attitudes towards ORS. The relationship between predictor factors on attitudes directly and indirectly towards this technique should be considered in the context of the future development of the ORS. This technique can be further improved through research and development and promoted by government agencies related to public health to control and prevent dengue disease in Malaysia.

## Author Contributions

**Conceptualization:** Latifah Amin.

**Data curation:** Ahmad Firdhaus Arham, Latifah Amin, Muhammad Adzran Che Mustapa.

**Formal analysis:** Ahmad Firdhaus Arham, Latifah Amin, Muhammad Adzran Che Mustapa.

**Funding acquisition:** Latifah Amin.

**Investigation:** Ahmad Firdhaus Arham, Latifah Amin, Zurina Mahadi.

**Methodology:** Ahmad Firdhaus Arham, Latifah Amin, Muhammad Adzran Che Mustapa, Zurina Mahadi.

**Project administration:** Ahmad Firdhaus Arham, Latifah Amin.

**Resources:** Latifah Amin, Zurina Mahadi, Mashitoh Yaacob.

**Supervision:** Latifah Amin, Zurina Mahadi.

**Validation:** Latifah Amin, Zurina Mahadi, Mashitoh Yaacob, Maznah Ibrahim.

**Writing – review & editing:** Ahmad Firdhaus Arham, Latifah Amin, Muhammad Adzran Che Mustapa.

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
