## [Decision Letter · Decision Letter 0]

30 Mar 2021

Dear Dr. Arham,

Thank you very much for submitting your manuscript "Stakeholder’s Attitudes in the Acceptance of Outdoor Residual Spraying Technique for Dengue Control in Klang Valley, Malaysia: a PLS-SEM Approach" for consideration at PLOS Neglected Tropical Diseases. As with all papers reviewed by the journal, your manuscript was reviewed by members of the editorial board and by several independent reviewers. The reviewers appreciated the attention to an important topic. Based on the reviews, we are likely to accept this manuscript for publication, providing that you modify the manuscript according to the review recommendations. 

Sincerely,

Scott B Halstead

Deputy Editor

Scott Halstead

Deputy Editor

Reviewer's Responses to Questions

**Key Review Criteria Required for Acceptance?**

**Methods**

-Are the objectives of the study clearly articulated with a clear testable hypothesis stated?

-Is the study design appropriate to address the stated objectives?

-Is the population clearly described and appropriate for the hypothesis being tested?

-Is the sample size sufficient to ensure adequate power to address the hypothesis being tested?

-Were correct statistical analysis used to support conclusions?

-Are there concerns about ethical or regulatory requirements being met?

Reviewer #1: The objective of the study is stated namely to determine the factors that influence the attitudes toward the Outdoor Residual Spraying (ORS)and the moderating factor. However, it needs to be consistent whether the authors try to investigate stakeholder’s attitudes or Malaysian’s attitudes. The study design needs to be improved. I think Figure 1 is not enough to present the conceptual research framework. It would be beneficial for the reader if the authors explained details regarding the exogenous variables, endogenous variables, a mediator variable, a moderator variable on the theoretical framework. The population is clearly described and appropriate for the hypothesis being tested. The author also has calculated the statistical power and showed that the sample size of respondents is sufficient. The statistical analysis used to support conclusions were correct. No ethical approval was required in this study.

**Results**

-Does the analysis presented match the analysis plan?

-Are the results clearly and completely presented?

-Are the figures (Tables, Images) of sufficient quality for clarity?

Reviewer #1: The results are well presented but it needs to be improved. For example, in Table 2, the author may need to explain how they categorized high and moderate on the overall mean score of the factors. Some results need to be clarified (file attached). The tables are clearly presented.

**Conclusions**

-Are the conclusions supported by the data presented?

-Are the limitations of analysis clearly described?

-Do the authors discuss how these data can be helpful to advance our understanding of the topic under study?

-Is public health relevance addressed?

Reviewer #1: The conclusion is supported by the data presented. The author clearly described the limitations of the analysis.

**Editorial and Data Presentation Modifications?**

Reviewer #1: I recommend "Minor Revision"

**Summary and General Comments**

Reviewer #1: This study aims to present the factors that influence the stakeholder's attitudes toward the outdoor residual spraying technique. Overall, the manuscript is generally well-written. However, some points need to improved and clarified as follows: 

Page 3: Under introduction, …serotypes-DENV-1, DEN-2,… 

Should be …serotypes-DENV-1, DENV-2,…

Page 6: Under theoretical framework, … “towards the ORS perceived benefit,…..

Should be “towards the ORS, perceived benefit,….. (please put comma)

Page 7, paragraph 1: “ Fig 1 presents …..Malaysians’ attitude towards……”

Should not be “ Fig 1 presents …..stakeholder’s attitude towards……”?Please be consistent!

Page 7: under “Trust in Key Players”: …. Benefits and risks associated with GM product…”

Please define the GM for the first time mentioned. Is that “Genetically Modified”?

Page 11: “In summary, the respondents were split into two groups: the public (n=197) and the scientists and implementers (n=202)” 

However, in the abstract, it is stated that “ …two stakeholder groups which consist of scientists, and the public ….”. 

Please be consistent in defining what the stakeholder is!

Page 12 under Results and Discussion: “ Table 2 present the overall mean scores of Malaysian attitudes…” Please be consistent!

Page 17 under Analysis of the Structural Model: “ The value of the standardized root mean square (SRMR) …”

Should not be ”The value of the standardized root mean square residual (SRMR) …”?

Page 18, the second paragraph: “ The Q2 values for the perceived benefit (0.120), perceived risk (0.069), and attitudes towards the ORS approach (0.287)…”

How did you get these numbers in bracket?

Should not be:

“The Q2 values for the perceived benefit (0.109), perceived risk (0.161), and attitudes towards the ORS approach (0.302)…”?

Page 21: ”… the VAF value was at 61.54%.”. How did you get 61.54?

Page 21, the last line: “…with the VAF value at 27.12%.”. How did you get 27.12%?

Page 23 under conclusion: “ The In conclusion…”. Please remove “The”.

PLOS authors have the option to publish the peer review history of their article (what does this mean?). If published, this will include your full peer review and any attached files.

Reviewer #1: No

Figure Files:

Data Requirements:

Reproducibility:

References

---

## [Editor Report · Decision Letter 1]

12 May 2021

Dear Dr. Arham,

Thank you very much for submitting your manuscript "Stakeholders’ Attitudes to Outdoor Residual Spraying Technique for Dengue Control in Malaysia: a PLS-SEM Approach" for consideration at PLOS Neglected Tropical Diseases. As with all papers reviewed by the journal, your manuscript was reviewed by members of the editorial board and by several independent reviewers. The reviewers appreciated the attention to an important topic. Based on the reviews, we are likely to accept this manuscript for publication, providing that you modify the manuscript according to the review recommendations. 

Sincerely,

Scott B Halstead

Deputy Editor

Scott Halstead

Deputy Editor

Figure Files:

Data Requirements:

Reproducibility:

References

---

## [Editor Report · Decision Letter 2]

21 May 2021

Dear Dr. Arham,

We are pleased to inform you that your manuscript 'Stakeholders’ Attitudes to Outdoor Residual Spraying Technique for Dengue Control in Malaysia: a PLS-SEM Approach' has been provisionally accepted for publication in PLOS Neglected Tropical Diseases.

Best regards,

Scott B Halstead

Deputy Editor

Scott Halstead

Deputy Editor

---

## [Editor Report · Acceptance letter]

24 Jun 2021

Dear Dr. Arham,

We are delighted to inform you that your manuscript, " Stakeholders’ Attitudes to Outdoor Residual Spraying Technique for Dengue Control in Malaysia: a PLS-SEM Approach ," has been formally accepted for publication in PLOS Neglected Tropical Diseases.

Best regards,

Shaden Kamhawi

co-Editor-in-Chief

Paul Brindley

co-Editor-in-Chief
